# Assessment of Cultural Ecosystem Services and Well-Being: Testing a Method for Evaluating Natural Environment and Contact Types in the Harku Municipality, Estonia

Fiona Nevzati [1,*], Mart Külvik [1], Joanna Storie [1], Liisa-Maria Tiidu [1] and Simon Bell [1,2]

1   Institute of Agricultural and Environmental Sciences, Estonian University of Life Sciences, 51014 Tartu, Estonia; mart.kylvik@emu.ee (M.K.); joannatamar.storie@emu.ee (J.S.); tiidu.liisamaria@gmail.com (L.-M.T.); simon.bell@emu.ee (S.B.)
2   OPENspace Research Centre, Edinburgh College of Art, University of Edinburgh, Edinburgh EH3 9DF, UK
*   Correspondence: fiona.nevzati@emu.ee

**Abstract:** This study examined the evaluation of cultural ecosystem services (CESs) and their impact on well-being in peri-urban areas, using a case study in Harku municipality, Estonia. CESs, encompassing intangible factors such as emotions and values, are crucial for well-being but challenging to assess. To address this, a pilot method was developed, involving a typology of natural environment types (NETs) and contact types (CTs), assessed by a panel of local experts. The results revealed that "spiritual, historic, and symbolic" gardens exhibited a strong positive connection to well-being. Blue and green spaces offering physical activities and aesthetics were also highly rated. Surprisingly, cemeteries scored higher than expected. Agreement among experts varied, with "parks + sporting" showing near-perfect consensus and weaker agreement found in "parks + food production", "blue spaces + providing gathering places", and "green landscape elements + education", highlighting diverse expert perspectives in identifying suitable combinations of NETs and CTs. This study addresses research-to-practice gaps and methodological challenges in applying CESs within planning frameworks, providing valuable insights for managing and conserving services in peri-urban areas. By testing the proposed method, this research contributes to a better understanding of how CESs can be effectively integrated into planning processes, fostering sustainable well-being in peri-urbanised regions.

**Keywords:** ecosystem services; spatial planning and design; landscape architecture; peri-urbanisation; peri-urban areas; inter-rater reliability; planning frameworks; expert panel

## 1. Introduction

The relationship between ecosystems and human well-being in urban areas is of the utmost importance. Ecosystem services (ESs), which are the benefits obtained from ecosystems, play a crucial role in supporting human societies. These services encompass provisioning services (such as food, water, timber, and fibre), regulating services (such as climate and water quality regulation), supporting services (such as nutrient cycling and soil formation), and cultural services (including aesthetic and recreational benefits) [1].

Human well-being (WB) extends beyond traditional measures such as income or gross domestic product (GDP). It encompasses social, environmental, and human rights dimensions, along with economic aspects [2–6]. Recognising the fundamental components of a good life, including freedom, choice, health, good social relations, and security, is crucial [1]. Therefore, understanding and promoting well-being necessitates considering not only economic factors but also the impact of ecosystems on mental and physical health.

With more than half of the global population now residing in cities and urbanisation rapidly progressing in various regions, including Europe [7,8], the loss and deterioration of ecosystems is escalating due to global and local environmental changes. If land conversion to urban land cover continues at its current pace, Europe may lose 10–15% of its ecosystems

provision value by 2050 [9–11]. Biodiversity loss impacts the properties of ecosystems and the benefits humans derive from them [12,13].

The expansion of peri-urban areas, located between urban cores and sparsely populated rural regions, is the main driver of eventual urbanizing, being the first transformation of the countryside towards urban character [7]. These peri-urban zones undergo social, economic, and spatial changes, varying between older industrial and post-industrial countries and younger industrialising nations and much of the developing world [14–16]. Changes in urban ecosystem composition can lead to increased fragmentation, reduced species abundance and richness, and adverse effects on human well-being resulting in urbanisation and biodiversity alterations [17–22].

Ecosystem losses result in economic and insurance costs and impact a wide range of cultural and social values in the long term [23,24]. Green and blue spaces, collectively known as green and blue infrastructure (GBI), constitute ecosystems in spatial terms and encompass a variety of natural and semi-natural areas that provide ecosystem services and protect biodiversity [25]. Unfortunately, the loss and deterioration of ecosystems, including urban ones, has significant consequences for both biodiversity and human well-being [17–22]. Recognising the value of biodiversity and the crucial role it plays in supporting well-being, various organisations are working towards promoting the interlinkages between the two aspects [21,26,27]. Ecosystem modifications, as underscored by various authors [28–30], have profound consequences for biodiversity. Without proper strategies, these changes can worsen the negative impacts on human health and well-being while disturbing the delicate balance of the human–ecological relationship. Prioritising the conservation of biological diversity is crucial to ensure the long-term provision of ecosystem services and other survival of species [31].

Cultural ecosystem services (CESs), defined as the nonmaterial benefits derived from ecosystems through spiritual enrichment, recreation, and aesthetic experiences, have been identified as significant contributors to well-being [1]. Urban ecosystems, consisting of various green and blue spaces such as parks, gardens, forests, and bodies of water, provide opportunities for physical and mental health benefits, as well as aesthetic and sensory experiences [23,24,32]. The World Health Organization (WHO) emphasises that health is not merely the absence of disease but a state of complete physical, mental, and social well-being, highlighting the importance of ecosystems in supporting human health [33].

However, the application of CES research in decision-making processes, particularly in planning, has been limited [23,34,35]. Challenges persist, including the selection of appropriate indicators, a lack of consensus on suitable methods, and difficulties in integrating economic and social values [36,37]. As a result, the application of CESs in decision-making processes still exhibits research-to-practice gaps. Innovative approaches are needed to ensure responsible and sustainable use of this knowledge [38].

Conserving and restoring ecosystem services play a critical role in strengthening urban resilience against environmental and socio-economic challenges such as climate change, pollution, and population growth. Haase et al. [39] and Chan et al. [37] emphasize the significance of ecosystem services in urban contexts, highlighting their role in promoting well-being and sustainable ecological relationships.

A pivotal step towards these goals is the mapping and assessment of ecosystems and their services (MAES). This process [1,40–42] enables a comprehensive understanding of the benefits derived from ecosystem services such as clean air, food, and recreation. MAES serves as an indispensable tool for informed decision making, identifying areas in need of conservation and sustainable management practices.

Although cultural ecosystem services (CESs) may seem intangible and difficult to quantify, their study offers numerous advantages. CES research aids policymakers and managers in informed decision making for land use and management practices. It also provides insights into the nonmaterial benefits derived from nature, particularly in urbanised areas, benefiting local communities and decision makers. Understanding how CESs contribute to well-being empowers individuals, organisations, and policymakers to effectively manage and conserve these essential services [23,38,43–45].

To bridge the existing research gap regarding the underestimation of green and blue infrastructure (GBI) and ecosystem services in urban planning and decision making, our study focuses on assessing the relationship between cultural ecosystem services and human well-being. To achieve this objective, we developed and implemented a novel methodology that enables us to assess the well-being benefits derived from a range of nature-based and cultural-based interventions.

The assessment in this study focused on two aspects: (i) the natural environment types (NETs) found in Harku, which encompass green and blue spaces such as parks, gardens, the sea, and cemeteries, and (ii) the cultural ecosystem services they provide, which were categorised into contact types (CTs). Examples of CTs include stress relief, recreation, a sense of community, and aesthetic appreciation. By mapping these two aspects, the study aimed to assess the potential of different combinations of NETs and CTs in promoting well-being. The research focused on a case study of Harku municipality, an area neighbouring Tallinn, the capital of Estonia. A methodology was developed and pilot-tested by a panel of experts to answer the following questions:

Research question 1 (RQ1): Which specific combinations of NETs and CTs have the highest and lowest potential for promoting well-being?

Research question 2 (RQ2): What are the advantages and limitations of utilising this novel method in assessing the nature-contact relationship?

The study aimed to achieve the following objectives:

1. To assess the potential of various combinations of natural environment types (NETs) and contact types (CTs) in promoting the well-being of residents in Harku municipality. Additionally, to determine the level of agreement among the panel of experts in rating these combinations.
2. To identify specific combinations of NETs and CTs that exhibit a higher or lower potential for enhancing well-being.
3. To develop and test a novel methodology for assessing the relationship between cultural ecosystem services (CESs) and well-being and to evaluate the advantages and limitations of this method in the context of the study.

In the 1990s, following the collapse of the Soviet Union and Estonia's independence, rapid urbanisation occurred as rural areas became depopulated, and old industrial areas were transformed into sprawling suburbs [46]. Eastern European cities, including Tallinn, experienced the highest level of land consumption in Europe from 2006 to 2012 [47]. The increasing number of urban peripheries worldwide presents significant challenges to sustainable development planning [48,49].

While previous studies have focused on mapping and assessing green, and to some extent, blue networks, the understanding of well-being through CESs in this area remains limited. The green and blue spaces in Harku are not only utilised by its residents but also attract frequent visitors from Tallinn due to their accessibility and diverse offerings. Therefore, the well-being benefits of these spaces extend beyond the boundaries of Harku municipality itself. It is important to comprehend the significance of CESs for the well-being of both residents and visitors and to preserve biodiversity in peri-urban areas that support those benefits.

By understanding the intricate dynamics between CESs, well-being, and residents' preferences, our study aims to inform urban planning and decision-making processes. Ultimately, our findings can play a crucial role in promoting sustainable development and improving the quality of life in peri-urban areas such as Harku.

## 2. Materials and Methods

### 2.1. Case Study

The case study focused on Harku municipality (59°23′21.876″ N, 24°34′45.12″ E), located west of Tallinn, the capital of Estonia. Harku is bordered by the Gulf of Finland, Lake Harku, and Tallinn to the east, and neighbouring municipalities such as Saue, Keila, and Lääne-Harju (Figure 1).

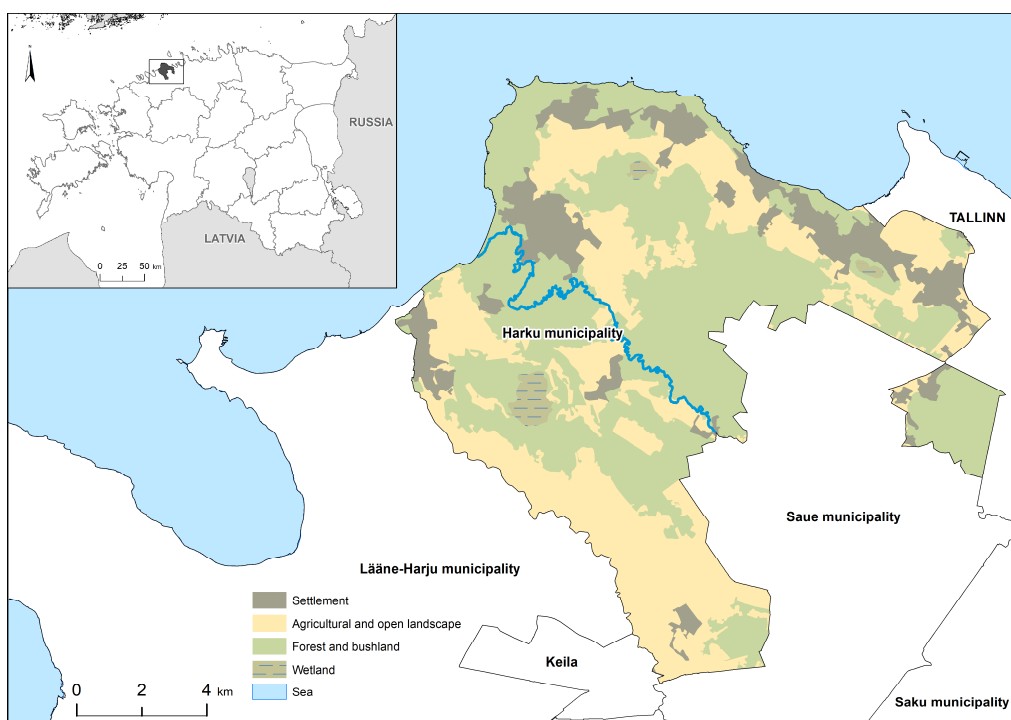

**Figure 1.** The location of the study area, Harku municipality, Estonia, and the main land-use types.

As of 1 December 2022, Harku had a population of 17,364 residents and covered an area of 159.7 km² [50]. The municipality has a forest cover of approximately 40%, with green areas predominantly located in the central part. There are also four nature conservation areas, several Natura 2000 sites, and other protected territories within the municipality [51].

Although the population density is lower than that of Tallinn, it is higher compared to most rural areas in Estonia [52]. The number of inhabitants fluctuates seasonally, being significantly higher during the summer. The study area can be considered to be peri-urban due to its location and characteristics. It exhibits a blend of urban and rural features, wetlands, and coastal areas (Figures 1 and 2). In the 1990s, over 20,000 Tallinn residents moved to the city suburbs. Higher social status individuals, about one-fifth of the total population, opted for prestigious housing in newly developed coastal locations such as Harku and Viimsi [53,54]. However, these areas face pressures from both urban and rural development, leading to conflicts between land uses and potential environmental degradation [55] as well as challenges in securing land for housing and agriculture [56].

Since Estonia regained independence in 1991, the country has experienced urban expansion, and it has observed a 33.96% increase in urban land in Harku since 1993 (Figure 2) [57]. As the country underwent economic and political transformations, the liberalisation of land ownership and the transition to a market economy led to increased urbanisation and population movement towards the fringes of the city.

The expansion notably accelerated from 2003 until the economic downturn in 2008. Despite a slight decrease in the overall population of Tallinn and its surrounding areas between 2000 and 2007, there was a significant period of robust and continuous housing construction in the suburbs of the city [53,54].

The aerial photos from 2002 and 2022 (Figure 2) depict two examples of the significant urbanisation transformation of Harku, including expanded housing construction, suburban growth, and encroachment into agricultural land. Estonia's EU accession in 2004 coincided with this period, providing opportunities and support for development, although not directly causing peri-urbanisation. Analysing the photos in the context of EU accession reveals the interplay between urbanisation, peri-urbanisation, and socio-political changes that shaped Harku's landscapes.

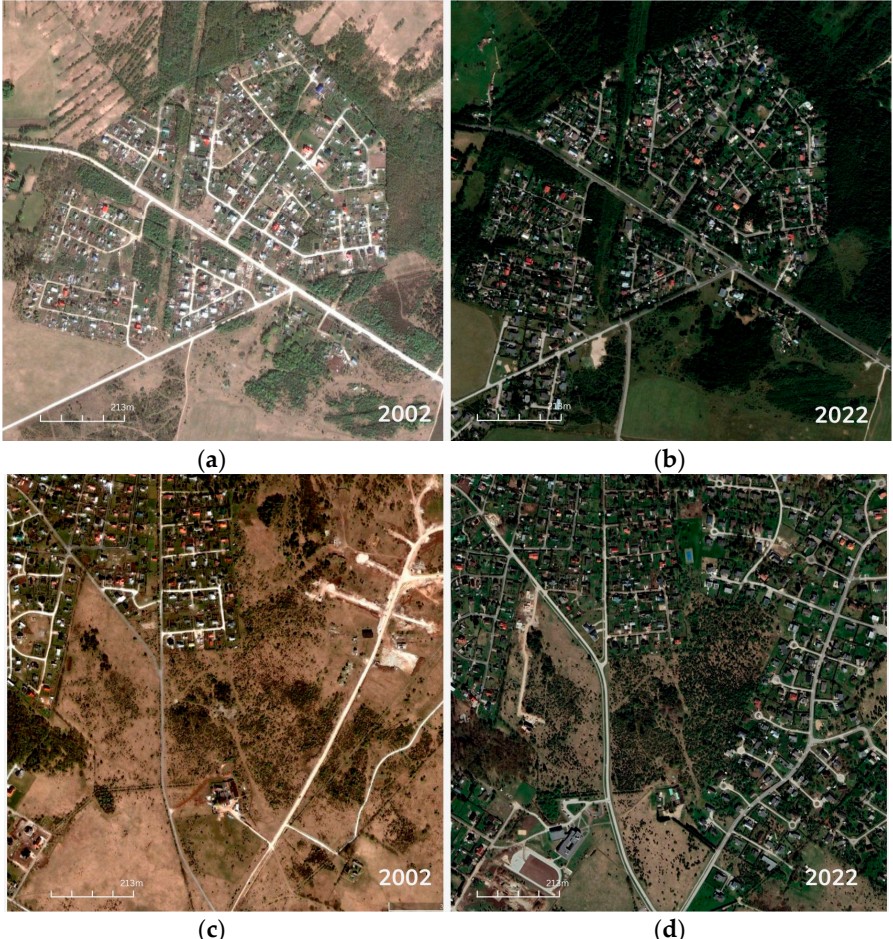

**Figure 2.** Aerial views of Harku show the urbanisation of the area in the past 20 years. (**a**,**b**) View 1: Harku in 2002 and 2022. (**c**,**d**) View 2: Harku in 2002 and 2022. Google Earth Pro, 2023 [58].

To better understand the relationship between the cultural ecosystem services (CESs) and well-being (WB) in Harku, our research strategy employed the following methods:

1. Development of a conceptual framework for assessing CES–WB linkages (explained in more detail in Section 2.2). This framework provides a theoretical basis for understanding how CESs contribute to WB and guides our analysis and interpretation of the study findings.

2. Application of a green and blue spaces preference method to evaluate the relative values of CESs and WB based on the perception of local experts (Section 2.3). This method allows us to gather insights into the preferences and priorities of experts regarding different NETs and CTs in promoting well-being. By using a matrix-based approach, we could capture and analyse their perceptions systematically.

These methods were chosen to investigate the specific combinations of the natural environment types (NETs) and contact types (CTs) that have the highest and lowest potential for enhancing well-being in Harku. Figure 3 shows examples of diverse landscapes, depicting a range of natural elements and human interactions (NETs and CTs) present in the case study area.

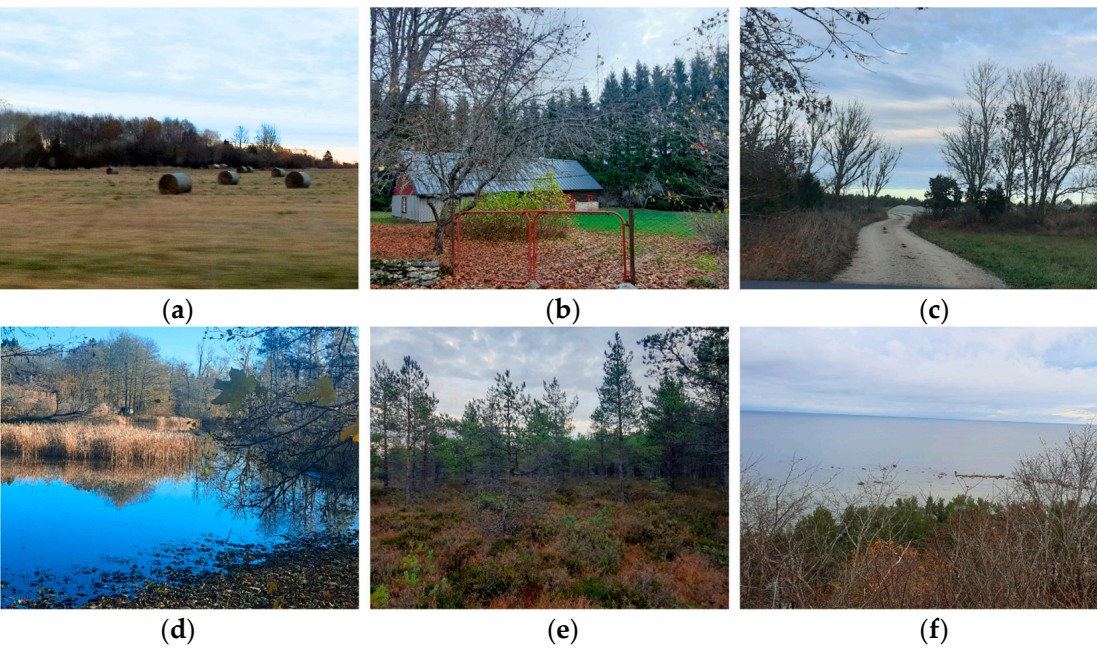

**Figure 3.** (**a**) agricultural land; (**b**) private house with a garden; (**c**) rural road; (**d**) a pond; (**e**) a forest; (**f**) a sea view from the Harku cliff.

Finally, we analysed the level of agreement among the panellists regarding these combinations (Section 3.1). Assessing the level of agreement or disagreement among experts provided valuable insights into the strength and reliability of the research findings, as well as the areas where their perspectives aligned or differed.

*2.2. Conceptual Framework*

Following the approach proposed by Hartig et al. [59], we developed a case-specific conceptual framework to illustrate the relationship between green and blue spaces, contact with nature, and well-being (Figures 4 and 5). This framework highlights the significance of contact with nature in influencing well-being and establishes the interconnected pathways between the natural environment, contact with nature, and well-being. While these pathways have traditionally been explored separately within different disciplines, our framework recognises their collaborative roles and interconnections [26,59].

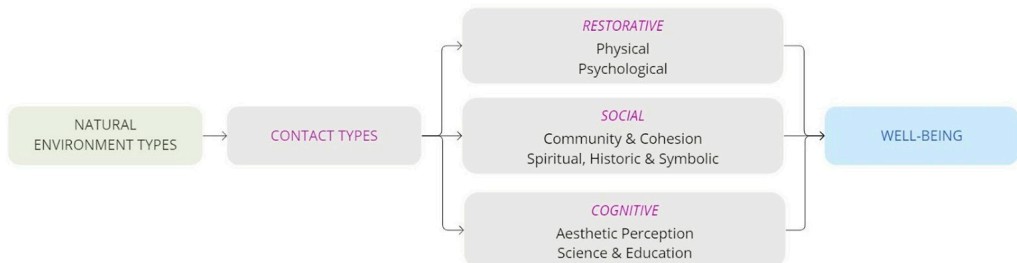

**Figure 4.** Adapted scheme from Hartig et al. [59].

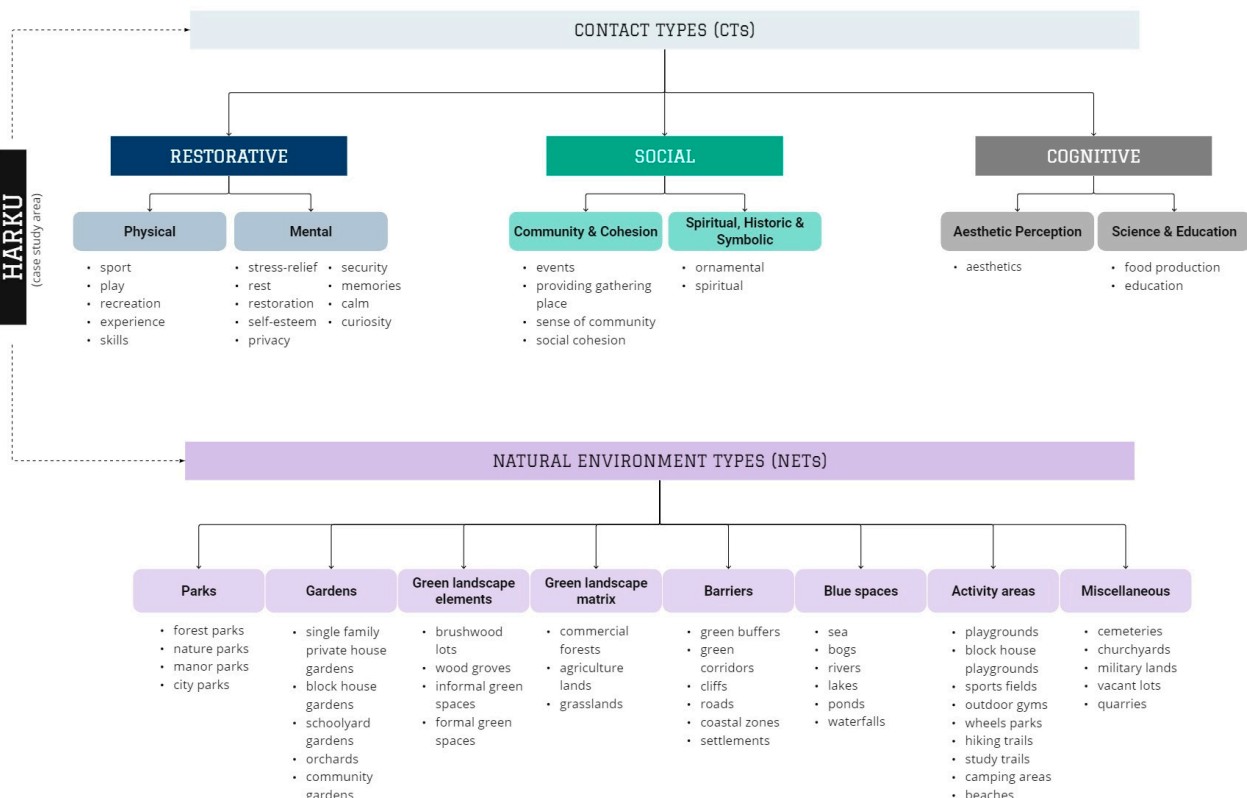

**Figure 5.** The conceptual framework.

Drawing from Hartig et al.'s adapted scheme derived in [59], we defined three distinct dimensions for assessment: (1) natural environment types (NETs), (2) contact types (CTs), and (3) well-being (WB) categories. This framework served as the foundation for evaluating these key aspects in our case study (Figure 4).

In our case study, we conducted an assessment focusing on two aspects: (i) natural environment types (NETs) and (ii) contact types (CTs). The NETs represent the green and blue spaces mapped within Harku. To classify the various green and blue space types, we referred to several sources such as Beatley and Newman [60] and Bell et al. [61]. The categorised typologies include (Figure 5):

- ○ *Parks* (forest, nature, manor, and city parks);
- ○ *Gardens* (single-family house private gardens, block house gardens, schoolyard gardens, orchards, community gardens);
- ○ *Green landscape elements* (brushwood lots, wood groves, informal and formal green spaces);
- ○ *Green landscape matrix* (commercial forests, agricultural lands, grasslands);
- ○ *Barriers* (green buffers and corridors, cliffs roads, coastal zones, settlements);
- ○ *Blue spaces* (sea, bogs, rivers, lakes, ponds, waterfalls);
- ○ *Activity areas* (playgrounds, sports fields, outdoor gyms, wheel parks, hiking and study trails, camping areas, beaches);
- ○ *Miscellaneous types* (cemeteries, churchyards, military lands, vacant lots, quarries) [62].

The next step involved identifying the possible contact types within these NETs, drawing from various sources on cultural ecosystem services (CESs) in green and blue spaces, such as Bishop et al. [63], Costanza et al. [64], Dai et al. [65], and the Millennium Ecosystem Assessment [1]. The contact types were categorised into three groups (Figure 5):

1. Restorative:
   - ○ *Physical* (sporting, playing, recreation, experience, and skills);
   - ○ *Psychological* (stress relief, resting, restoration, self-esteem, privacy, security, memories, calming, and curiosity).

2.    Social:

  ○    *Community and cohesion* (events, providing a gathering place, sense of community, and social cohesion);
  ○    *Spiritual, historic, and symbolic* (ornamental and spiritual).

3.    Cognitive:

  ○    Aesthetic perception (aesthetic);
  ○    Science and education (food production and education) [62].

### 2.3. Matrix

To evaluate the connectedness rates between the NETs and CTs, we developed a matrix within our conceptual framework. The matrix design was based on the approach proposed by Burkhard et al. [66], with CTs and NETs placed on the x- and y-axes, respectively (see Supplementary Materials for the matrix layout).

The initial version of the matrix was created and tested by a landscape architect with extensive local knowledge. This allowed us to refine and improve the matrix based on practical considerations and local context. The landscape architect provided valuable insights and suggestions for optimising the matrix structure and ensuring its relevance to the specific characteristics of Harku.

Subsequently, the revised matrix was further tested and validated by a panel of experts representing various relevant disciplines and expertise (see Supplementary Materials). The development and testing of the matrix allowed us to refine our research methodology and ensure its effectiveness in assessing the CES–WB linkages.

Overall, the matrix served as a valuable tool for systematically analysing and interpreting the data collected using the knowledge of local experts regarding the relationship between NETs and CTs. It enabled us to assess the strength and importance of different CES–WB connections and identify the combinations that have the highest potential for enhancing well-being and also provided a means to assess the level of agreement among the experts when scoring the NETs and CTs.

### 2.3.1. Expert Panel

Qualitative assessments of the relationship between NETs and CTs were conducted using a panel of local experts with diverse backgrounds and expertise. The panel consisted of a chief forest manager with in-depth knowledge of the local natural environment, a spatial developer familiar with the urban and rural landscape, a former council member and mayor with a deep understanding of community dynamics, a municipality architect with expertise in design and planning, and a local school teacher and geographer who provided insights into the educational and social aspects of the area (Table 1).

**Table 1.** Experts and their occupations.

| Expert | Occupation |
| --- | --- |
| Expert 1 | Spatial developer, outdoor sports activist |
| Expert 2 | Local school teacher, geographer, neighbourhood activist |
| Expert 3 | Chief state forest manager, heritage conservation activist |
| Expert 4 | Former municipality mayor and council member, environmental activist |
| Expert 5 | Municipality architect and planner |

The matrix (Supplementary Materials) was presented to the local expert panel members, providing detailed information about the different NETs and CTs. Before the assessment, the material was translated from English to Estonian to ensure a clear understanding for the experts.

To facilitate the assessment process, the field researcher conducted an extensive explanation of the various types of NETs and CTs as indicated in the matrix (Supplementary Materials). The researcher elaborated on the distinct characteristics and features of each type, enabling the experts to make informed assessments and ratings based on their professional knowledge and experience.

The panellists were requested to assess and rate the degree of connection between NETs and CTs on a scale ranging from 0 (indicating no connection) to 3 (indicating a high level of connection). Each expert evaluated the contact types sequentially, beginning with the first group of NETs, allowing for a systematic and comprehensive evaluation of the relationship between humans, natural environments, and their level of contact (Figure 6).

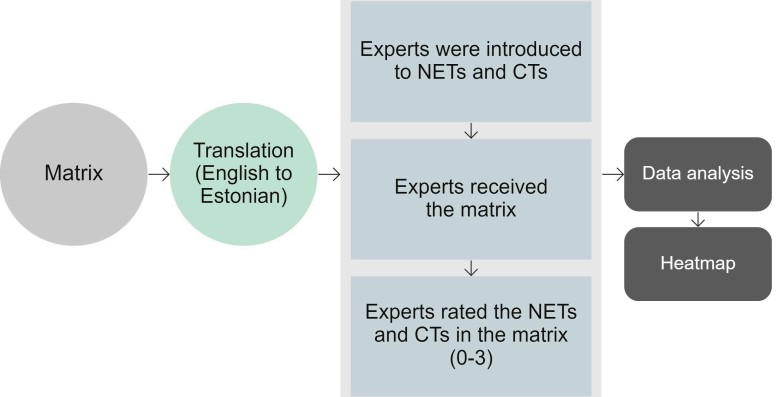

**Figure 6.** The expert panel process.

Following the individual assessments, a panel discussion was held to compare and discuss the ratings provided by each expert. The aim was to capture the collective expertise and insights of the panel members, ensuring a comprehensive evaluation of the relationship between green and blue spaces, contact types, and well-being.

Following the completion of the individual assessments, the ratings provided by each expert were subjected to further analysis using a statistical method to assess the inter-rater reliability. This approach aimed to evaluate the degree of agreement among the experts and provide a quantitative measure of the consistency in their evaluations.

To conduct the analysis, the ratings of each combination of NETs and CTs were compiled into a dataset. A statistical technique, Fleiss' Kappa, was employed to examine the level of agreement among the experts' ratings (see Section 2.3.2 for more detail). These coefficients provided a measure of the extent to which the experts' ratings aligned with each other.

### 2.3.2. Statistical Analysis—Fleiss' Kappa

By utilising a statistical method, we aimed to quantify objectively the inter-rater reliability of the expert panel and to provide an additional layer of rigour to the assessment process. This analysis allowed us to assess the consistency and agreement in the evaluations, contributing to the reliability and validity of the findings. The results of the statistical analysis, described in Section 3.1, provided valuable insights into the overall reliability of the expert-based assessments and informed the interpretation of the relationship between NETs, CTs, and well-being.

We assessed the agreement among the panel of experts in identifying the NETs and CTs for inter-rater agreement for categorical response variables following the approach of Fleiss [67] and Fleiss et al. [68]. This analysis was conducted in Excel 2016 where we represented each combination of NETs and CTs along with the ratings provided by the panellists (ranging from 0 to 3) and calculated the Fleiss' Kappa score to determine the strength of agreement among the panellists.

In addition to the inter-rater reliability analysis, we also conducted an analysis of the average mean scores (described in Section 3.2) that addressed our first research question. The average mean scores were calculated by aggregating the ratings provided by the expert panel for each combination of NETs with CTs.

By analysing the average mean scores, we aimed to explore the overall patterns and tendencies in the experts' assessments. This analysis allowed us to identify the NETs and CTs that were consistently rated higher or lower in terms of their perceived connections and impact on well-being. By considering the average mean scores, we gained insights into the relative importance and contribution of different green and blue spaces and contact types to well-being in the study area.

The analysis of the average mean scores complemented the inter-rater reliability analysis, providing a broader perspective on the relationship between green and blue spaces, contact types, and well-being. It helped us address our research question 1, assessing the degree of connection between NETs and CTs in promoting well-being.

## 3. Results

### 3.1. Analysis of Panel Agreement—Fleiss' Kappa

The results of the analysis showed the level of agreement among the panel of experts in identifying natural environment types (NETs) and contact types (CTs). The Kappa value was then classified as indicating weak or strong agreement, following the criteria outlined by Landis and Koch [69].

The Fleiss' Kappa classification has six levels and a kappa range from 0.00 poor to 1.00 almost perfect [69]. In this study, the agreement levels, as measured by Kappa values, varied across different combinations of NETs and CTs. The strongest group of agreements (Figure 7) showed Kappa values ranging from 0.70, indicating substantial agreement, to 1.00, representing almost perfect agreement. On the other hand, the weakest group of agreement had Kappa values ranging from 0.18 (slight) to 0.21 (fair) (Figure 7).

The highest agreement was observed for the combination of "parks + sporting", with a Kappa value of 1.00, indicating an almost perfect level of agreement among the panel of experts. The next highest level of agreement was observed for "gardens + ornamental" with a Kappa value of 0.92, which falls under the category of substantial agreement (Figure 8).

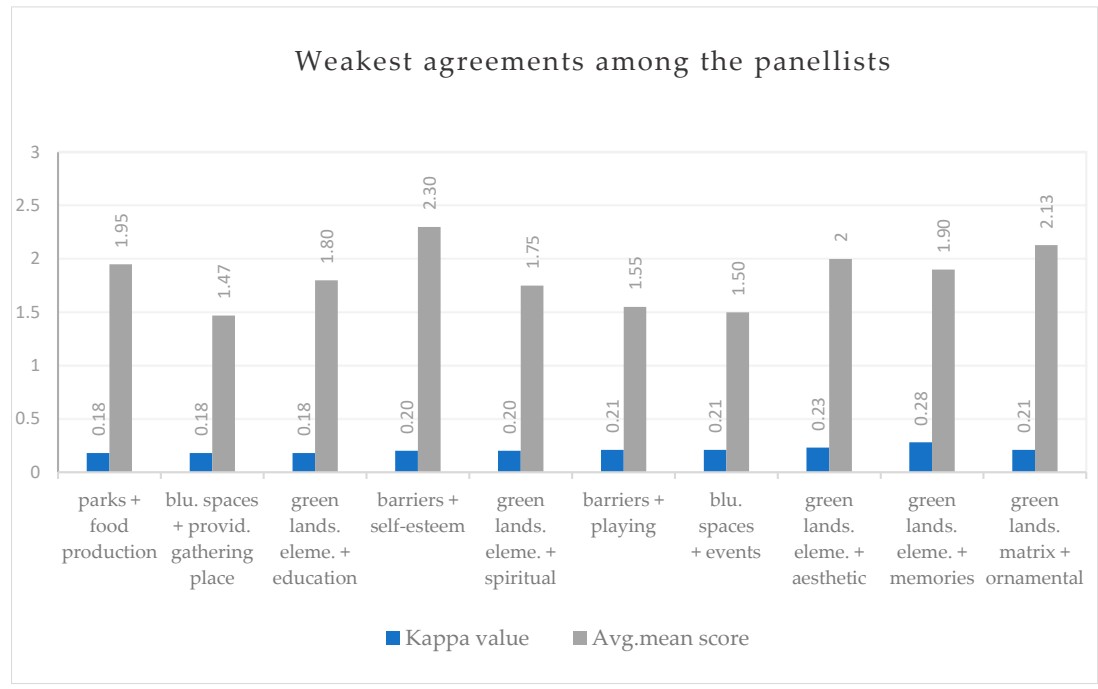

**Figure 7.** Weakest agreements among the panellists.

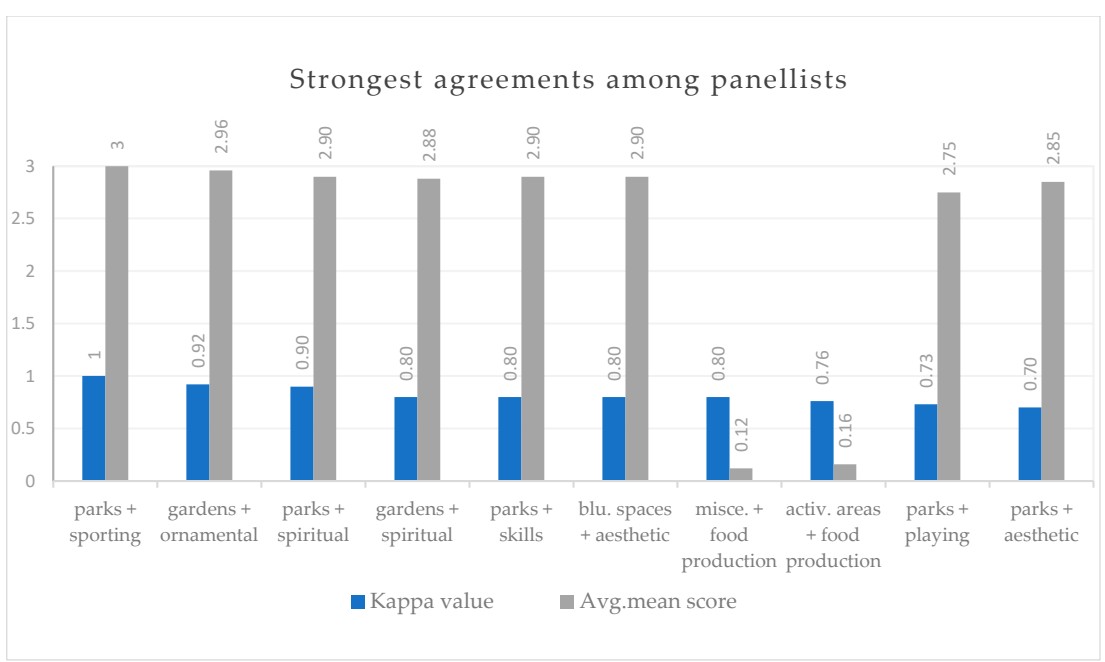

**Figure 8.** Strongest agreements among the panellists.

The combination of "parks + spiritual" and "gardens + spiritual" showed a Kappa value of 0.90 and 0.80, respectively, indicating a substantial and fair level of agreement among the panellists. Similarly, the combinations of "parks + skills", "blue spaces + aesthetic", "miscellaneous + food production", and "activity areas + food production" showed Kappa values ranging from 0.76 to 0.80, which fall under the category of moderate to substantial agreement. On the other hand, the combinations of "parks + playing" and "parks + aesthetic" showed Kappa values of 0.73 and 0.70, respectively, indicating only a moderate level of agreement among the panel of experts (Figure 8).

The weakest agreements were found in the ratings of "parks + food production", "blue spaces + providing gathering places", and "green landscape elements + education", all with a Kappa value of 0.18, indicating only a slight level of agreement. Furthermore, the ratings of "barriers + self-esteem", "green landscape elements + spiritual", "barriers + playing", "blue spaces + events", and "green landscape matrix + ornamental" showed slightly higher but still weak levels of agreement ($\kappa = 0.20$–$0.21$). Lastly, the combinations of "green landscape elements + aesthetics" and "green landscape elements + memories", with $\kappa = 0.23$–$0.28$, indicated a relatively higher level of consensus compared to the weakest agreement, but still fall within the category of low agreement (fair) (Figure 7).

### 3.2. Analysis of Panellists' Ratings—Average Mean Score

In the second part of our analysis, we looked at the average mean scores for the different NETs and CTs based on the ratings given by the panellists (0–3). We grouped the ratings into the main categories for CTs (physical, psychological, community and cohesion, spiritual, historic and symbolic, aesthetic perception, and science and education) and NETs (parks, gardens, green landscape elements, green landscape matrix, barriers, blue spaces, activity areas, and miscellaneous). This helped us to obtain a better understanding of the panellists' preferences and perceptions regarding well-being. We then identified the NETs and CTs with the highest and lowest average scores, which allowed us to address research question 1.

According to the average mean score, in terms of NETs scores, gardens ranked highest (2.63) with the subsection schoolyard gardens the highest of these; parks were the second highest (2.60) with nature parks the highest in this section. Blue spaces, which include areas like lakes or the sea, ranked as second highest category with an average score of 2.60.

Among blue spaces, the sea had the highest score (2.21). The lowest scores (1.29) were in the miscellaneous types group, with quarries and vacant lots scoring the lowest (0.53). The highest ranked CT category was "aesthetic" (2.49), strongest in blue spaces (2.90). "Psychological" (2.22) was strongest in parks and gardens (2.66), and "spiritual, historic, and symbolic" (2.19) was strongest in gardens (2.92).

In the following subsections, we present the results of each CT and examine how they relate to the NETs based on the average mean scores.

### 3.2.1. Restorative—Physical

Parks had the strongest association (with an average mean score of 2.86), while miscellaneous types of NETs had the weakest (0.69). Within the physical subcategory, nature parks were the only green spaces that received a high score of 3 for "playing". In contrast, most blue spaces such as bogs (1.60), rivers (1.80), ponds (1.20), and waterfalls (1) had low scores. However, exceptions to this were the sea (2.80) and lakes (2), which had relatively higher scores. Blue spaces such as the sea, bogs, waterfalls, and beaches (from the activity areas NET) had the highest (3) score for "recreation". Forest parks, nature parks, and beaches received the highest score (3) for "experience". In the "skills" category, only forest parks and nature parks scored the highest (3), while the lowest NETs were found in the miscellaneous types (0.7) (Figure 9).

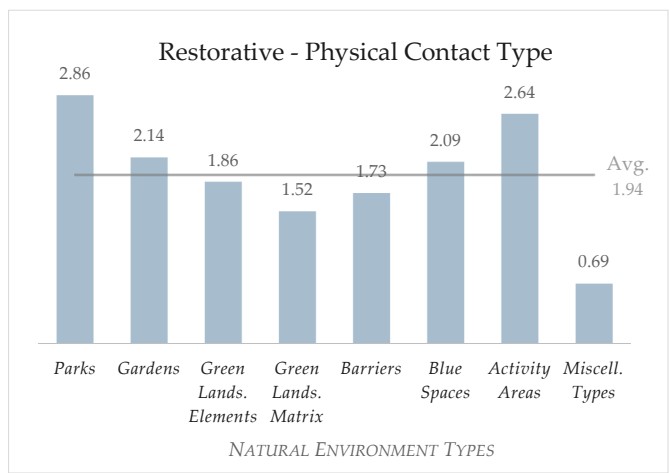

**Figure 9.** Restorative—Physical CTs + NETs results—average mean scores.

### 3.2.2. Restorative—Psychological

The highest scores (2.66) were given to parks and gardens, and the lowest (1.35) to miscellaneous types of NETs. The only two high scores (3) for "stress relief" were beaches and schoolyard gardens, and the overall average score was 2.22.

Among the NETs subgroups, the sea, bogs, and beaches received the highest score of 3 for "resting". Forest parks and beaches were the top choices for "restoration" with a score of 3. However, there was no high score for "self-esteem", although most NETs scored moderately high at 2.40. The lowest rated for "privacy" were the quarries (0) followed by agricultural land (0.80). Blockhouse playgrounds were moderately low (1.8) under this CT. The highest for "privacy" were the single-family house, private gardens, and forest parks. However, when the two were compared for "security", the forest parks scored lower (2.40) but the single-family house private gardens had the maximum score (3). "Memories" were strongest for the beaches, and cemeteries (3). The only maximum score (3) for "calming" was given to "beaches". Generally, however, parks, gardens, blue spaces, and activity areas scored moderately highly (2.40) for "calming". The only high score (3) for "curiosity" was for the beaches, whereas ponds were the only moderately low-scored NET from all the rest of the blue spaces. The rest of the NETs scored moderately high for "curiosity", including cemeteries, churchyards, and military land (Figure 10).

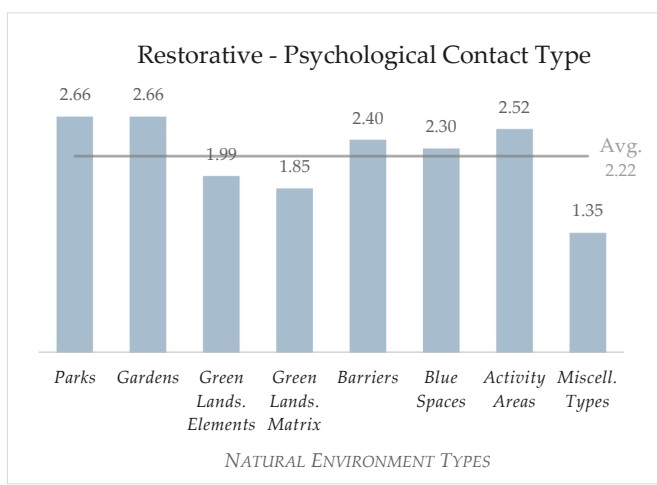

**Figure 10.** Restorative—Psychological CT + NETs results—Average mean scores.

### 3.2.3. Social—Community and Cohesion

With an average score of 1.91, the highest scores in the "community and cohesion" NETs group were for gardens (2.63), with the lowest for the green landscape matrix (1.30). The two high scores (3) for the subgroup "events" were the settlements and the beach, with gardens, parks, and activity areas the second highest. The top three maximum scores (3) for "providing a gathering place" were given to the beaches, settlements, and community gardens. Vacant lots were given a higher score (1.20) than the pond (0.80) for "providing a gathering place". The blue spaces were generally low (1.60) in comparison with gardens (2.70) and parks (2.50) for the "sense of community" CT—even the cemeteries and churchyards scored higher (2.60) than the blue spaces. The highest-scored (3) for "social cohesion" were settlements, schoolyard gardens, community gardens, and beaches (Figure 11).

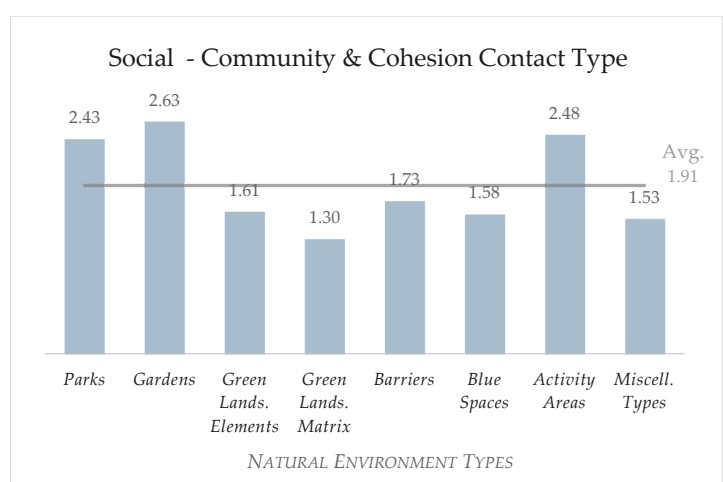

**Figure 11.** Social—Community and Cohesion CT + NETs results—average mean scores.

### 3.2.4. Social—Spiritual, Historic, and Symbolic

With an average score of 2.28, the highest scores were for the gardens (2.92), the lowest for the miscellaneous (1.70). Within the subgroups, the highest under "ornamental" were manor parks, single-family private gardens, block house gardens, schoolyard gardens, community gardens, the sea, lakes, beaches, cemeteries, and churchyards. The "spiritual" CT gave similar results including forest, manor, and nature parks (3). The lowest in "ornamental" were green landscape elements and miscellaneous types (1.80), and in "spiritual", miscellaneous (1.60) (Figure 12).

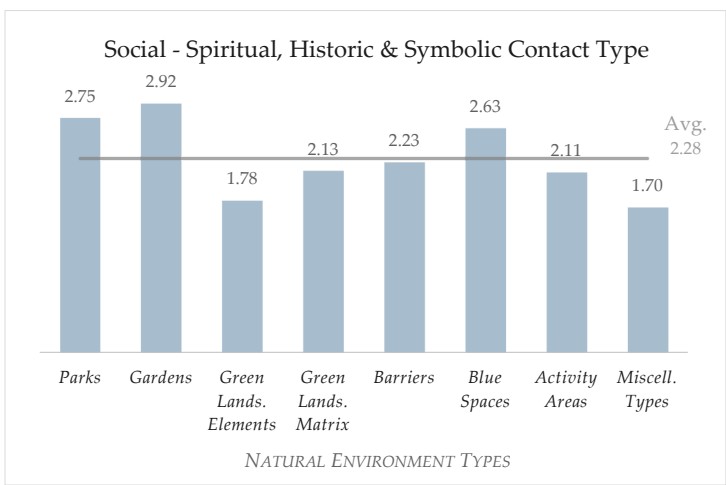

**Figure 12.** Social—Spiritual, Historic, and Symbolic CT + NETs results—average mean scores.

### 3.2.5. Cognitive—Aesthetic Perception

The highest scores in this category were for blue spaces (2.90), parks (2.85), and gardens (2.84), and the lowest for miscellaneous types (1.60). The "aesthetic perception" CT gave relatively high results in every NET with a 2.49 score on average. Regarding NETs subgroups, the highest scores (3) were seen in manor parks, orchards, commercial forests, coastal zones including cliffs, the sea, rivers, waterfalls, beaches, cemeteries, and churchyards. The lowest scores were for vacant lots (0.80) and quarries (0.20) (Figure 13).

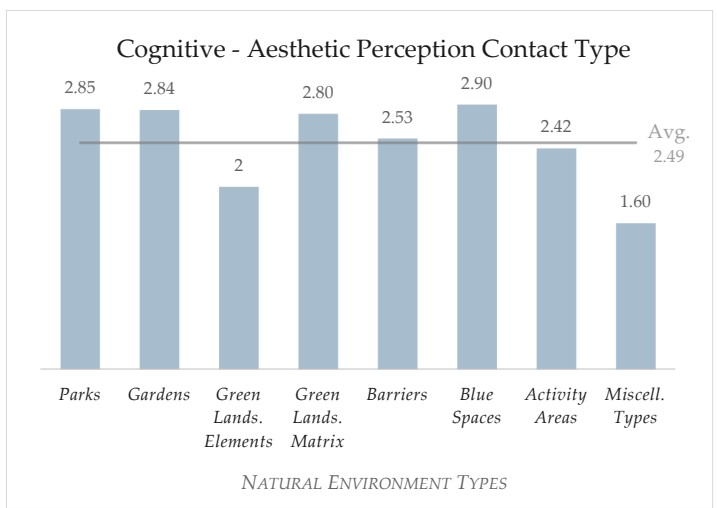

**Figure 13.** Cognitive—Aesthetic Perception CT + NETs results—average mean scores.

### 3.2.6. Cognitive—Science and Education

The highest score (2.60) was for gardens, the lowest for miscellaneous types (0.88), with an overall average of 1.66. The subgroup "education" had two maximum scores (3), from schoolyard gardens and study trails, and "food production" had only one maximum score (3) under orchards, and slightly lower (2.80) for community gardens (Figure 14).

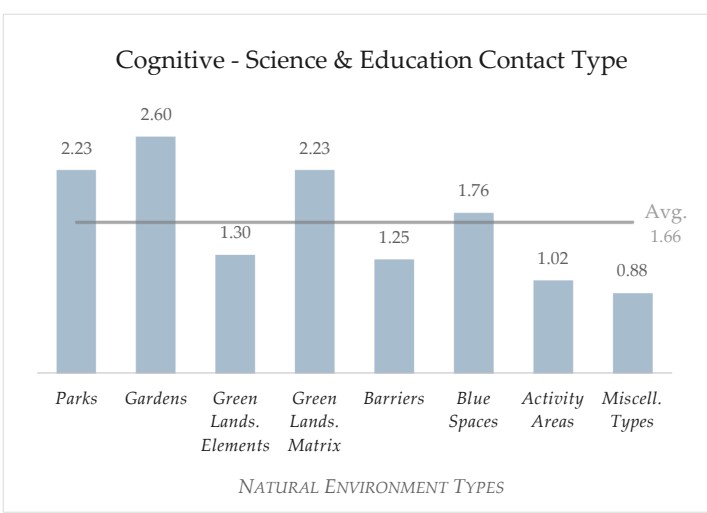

**Figure 14.** Cognitive—Science and Education CT + NETs results—average mean scores.

## 4. Discussion

### 4.1. Analysis of Panel Agreement—Fleiss' Kappa

The results of our study showed that the panel of experts had a moderate to substantial level of agreement in identifying the natural environment and contact types but some combinations did show a low level of agreement. The highest agreement was observed for the combination of "parks + sporting", whereas the lowest level of agreement was observed for "parks + food production", "blue spaces + providing gathering places", and "green landscape elements + education".

Notably, all types of parks, including forest, nature, manor, and city parks were perceived as highly beneficial for sporting activities by the panellists. This indicates that parks are viewed positively as places that offer opportunities for physical activities and recreation. This finding is particularly relevant given the well-documented benefits of physical activity for health and well-being [70–73].

In contrast, the findings revealed that the panellists had the weakest agreement when it came to rating green landscape elements with "education", blue spaces with "providing gathering places", and parks with "food production". This suggests that there is a higher degree of variability in how these types of natural environments are perceived and valued by experts. It may also suggest weak general traditional associations, for example, it is only recently that parks may include community gardens within their borders [74].

The disparity in agreement regarding green landscape elements with "education" signals the need for additional exploration and research in this area. Understanding education in natural environments is a complex and multifaceted concept, and the panellists' perceptions are likely shaped by various factors, including the level of education, age groups of the users, and the types of natural environments.

Similarly, the lack of consensus surrounding blue spaces in terms of "providing gathering places" and parks with "food production" underscores the importance of further investigation into these aspects of natural environments. The panellists' perspectives may be influenced by their individual experiences, preferences, and broader cultural and societal factors. Hence, it is vital to incorporate diverse viewpoints when designing and managing green and blue spaces.

The moderate level of agreement among the panellists suggests that while there is some consensus on the benefits of natural environments and the types of contact they offer for well-being, there is also significant variability in the perception of NETs and CTs among experts. This is not unexpected, as the main aim of this study was to test a method for evaluating the preferences and perceptions of natural environments and types of contacts, rather than to achieve complete consensus among the panellists. Our goal was

to demonstrate the feasibility of using a systematic approach to assess the different NETs and CTs they offer. In this regard, we have achieved our objective.

However, the variability in expert opinions also highlights the importance of considering diverse perspectives and multidisciplinary teams when designing and managing natural environments. This inclusivity becomes particularly important in light of potential conflicts arising from different perceptions of the landscape and societal values. These conflicts can give rise to tensions surrounding the appropriate use and management of natural environments [75,76]. Therefore, it is imperative to explore and develop methods that effectively evaluate and integrate these diverse viewpoints. By doing so, we can foster an inclusive environment and a more balanced approach that encompasses the well-being of both individuals and society as a whole.

### 4.2. Analysis of Panellists Ratings—Average Mean Score

In response to research question 1 (RQ1), which aimed to identify the specific combinations of natural environment types (NETs) and contact types (CTs) that have the highest and lowest potential for promoting well-being in the Harku municipality, our findings revealed that certain combinations of NETs and CTs can effectively promote well-being. However, some combinations were less effective than others.

The study showed that aesthetic appreciation of green and blue spaces was the highest-ranking contact type (CT) among the local experts, which indicates an appreciation for aesthetically pleasing green and blue spaces [77]. As previous studies show, aesthetic experiences of green space in high biodiversity areas are valued more than those in low biodiversity areas. Therefore, urban greenery is perceived as having the greatest positive value where biodiversity is abundant [78]. To design settings that serve both nature and people, landscape architects need to understand the connections between biodiversity and well-being [79].

Furthermore, the results of our study showed a high score for "aesthetic" CT in connection to bogs. In Estonia, bogs cover about 7% of the land, and the oldest bogs date back more than ten thousand years [80]. This means that Estonians have watched bogs and forests grow for thousands of years and have developed their culture alongside them, furthermore, bog pools in Estonia are numerous, with a total of 45,309 bogs [81] and actively used for swimming, walking, and resting, thus, bogs showed high scores in "ornamental", "spiritual", "stress-relief", and "education".

Bogs, forests, cemeteries, and schoolyard gardens were also rated highly under different subgroups of CTs, with forests being particularly valued due to their cultural significance since the forest cover in Harku is up to 40% and more than 50% in the whole of Estonia [51]. Furthermore, they are one of the most valuable economic and cultural resources of the country [82–84].

High scores were also seen in cemeteries, so they are not only burial places but also serve other purposes, especially in cities. Aside from providing ecological and recreational benefits, their general use plays a significant role in local communities. Moreover, over the centuries, Estonian cemeteries have been built close to large trees or within forest stands; thus, these public green spaces often have a variety of tree species [85]. We found that cemeteries together with churchyards invoke feelings of safety and privacy, a strong sense of social cohesion, and memories, all of which contribute to their overall well-being benefits which are in line with previous findings [85].

Schoolyard gardens were rated highest for their ability to reduce stress, promote social cohesion, add ornamental value, be spiritually meaningful, and provide educational benefits. Previous studies have also reached similar conclusions to illustrate the benefits of these types of benefits for promoting well-being in students and strengthening the school as a positive environment for youth development [86–88].

It is interesting to note that single-family private gardens, block house gardens, and schoolyard gardens had a maximum score in "spirituality"; however, the exact significance of this relationship is unclear due to the lack of evidence between gardens and spirituality

and exactly how spirituality was understood by the panellists. Alternatively, this could mean that the garden is seen as a spiritual place and gardening is considered a spiritual activity or a spiritual journey [89] in addition to being rated highly in our study for its calming effects.

In terms of blue spaces, the sea was rated the highest of all NETs. The sea was also found to be highly valued for its many possibilities for activities for the inhabitants and visitors, but people often feel threatened by the risk of injury (such as the Harku cliff), indicating a low score in the "security" subcategory. However, the cliffs have been found to be aesthetically valuable. When in a stressful or overwhelming situation, people tend to visit particular blue places where they can listen to the relaxing sounds of aquatic environments [90], bringing them restorative benefits and a sense of relaxation. A similar pattern was found between the sea and bogs, promoting rest, stress relief, restoration, and calm. Presumably, this is connected to the sound of waves and the stillness of the bogs.

Parks were found to provide physical, social, and mental benefits, with nature parks ranking the highest for "playing", "experience", "skills", and "spirituality" as in previous studies [59,73,91–94]. However, parks were also considered communal and not private spaces, and forest parks scored low under the "security" CT.

Agricultural land was the third lowest rated NET, after vacant lots and quarries. More than a third of Estonia's population lives in rural areas, and less than nine per cent are employed in agriculture [46,95,96].

Quarries and vacant lots are the least preferred places as they are not usually places people would choose to go for a sense of well-being. They are often considered places that have an unpleasant character and could be dangerous and usually left unmanaged, which creates uncomfortable feelings in people (fear, lack of security and safety, fear of the unknown, etc.) [62].

It is important to engage residents in identifying local health impacts and generating solutions to vacant lots for them to accept and apply community-based solutions [97]. Moreover, this supports social and environmental sustainability by promoting the multi-purpose use of spaces, e.g., using vacant land for urban agriculture [49,98]. However, it is important to note that the benefits of the natural environment are complex and multidimensional, and can vary according to individuals' characteristics, such as age, gender, and socio-economic status [99]. Furthermore, cultural and contextual factors may influence the perceived benefits of different NETs and CTs [100].

In light of the growing interest in the relationship between green spaces and well-being, there is a need for further investigation into several areas, specifically the relationship between biodiversity and well-being, particularly in peri-urban green spaces at different levels of biodiversity and concerning different aspects of well-being [101,102], such as social–ecological biodiversity, focusing on the relationship between biodiversity and human activities [103], mental health [104], and subjective well-being [105]. This is especially important in the design of landscapes that enhance multiple ecosystem services [106].

Another area that warrants further exploration is the spiritual and cultural significance of gardens and green spaces. By conducting qualitative research, we can gain a deeper understanding of how people perceive spirituality in these spaces and how it contributes to their well-being [74]. De Lacy and Shackleton [107] analyse how urban green spaces offer aesthetic and spiritual ecosystem services, including a sense of connection with nature and transcendental experiences. Moreover, acknowledging and valuing these services is crucial for enhancing well-being and conserving urban ecosystems [107] and peri-urban areas [108].

Cemeteries are often overlooked as public green spaces, despite their potential impact on community well-being and providing cultural ecosystem services [109], as also shown in our study. Conducting qualitative research on how people perceive and use cemeteries can help to shed light on their role in promoting well-being [87] and biodiversity conservation [110]. The natural forest environment in which the majority of Estonian cemeteries are located provides a serene and natural setting, making this context particularly interesting for further research. However, interventions and interpretations in cemeteries must

be carefully planned to balance respect and curiosity [111]. Moreover, according to our study, there is a need to explore the potential benefits of agricultural land for well-being, particularly in rural areas, to understand how it can promote well-being in communities.

Regarding research question 2, we found that utilising a novel method in a pilot study assessing a nature-contact relationship had both advantages and limitations. One major advantage of our approach was that it provided a comprehensive toolkit for landscape analysis. The green and blue spaces preference method enabled us to determine the perceived values of cultural ecosystem services and well-being as reported by the local experts in the matrix. Our study identified significant gaps between research and practice when applying the cultural ecosystem services framework and highlighted the need for more suitable methods to evaluate these approaches. This is visible in many other studies that have found similar challenges in translating research findings into practice. While our study provided valuable insights into this issue and a framework for a method, it is important to acknowledge its limitations. It is important to acknowledge that our sample size was limited as our focus was on testing a novel method. To enhance generalisability, future studies should include a larger and more diverse group of experts with varying backgrounds and perspectives. However, this might be challenging in rural landscapes, especially in low-density countries such as Estonia where experts are fewer in number. Efforts should be made to include as diverse a group as possible within these limitations.

Despite these limitations, we believe that our study contributes significantly to the field of landscape architecture and ecosystem services (specifically cultural) research highlighting the need for more research to understand better the mechanisms through which natural environments promote well-being in peri-urbanised areas. Furthermore, our approach could provide a strong foundation for future studies involving more diverse populations and larger sample sizes. Moreover, incorporating additional parameters such as "duration" and "quantity" of time spent in green and blue spaces provides a more profound understanding of their contributions to well-being.

To maximise the impact of research going forward, it is crucial to involve subject matter experts and stakeholders. By collaborating with them and utilising innovative methods such as AI-based tools, such as UrbanistAI [112], researchers can gather valuable insights and generate customisable designs that align with stakeholder preferences. To ensure interventions in peri-urban areas are relevant and effective, researchers should adopt a participatory approach, consulting experts, engaging stakeholders, and exploring technological solutions.

## 5. Conclusions

This study investigated how nature affects well-being in the peri-urban area of Harku, in northern Estonia. We used two methods, creating a framework to evaluate the relationships between cultural ecosystem services and well-being based on two aspects: natural environment types (NETs) and contact types (CTs). NETs refer to the green and blue spaces in Harku municipality, while CTs represent the potential cultural ecosystem services in the area. Furthermore, we utilised a green and blue spaces preference method to determine the perceived values of cultural ecosystem services and well-being, as reported by local experts in the matrix.

In answering research question 1 (RQ1), we found that certain CTs rate higher than others, more precisely, the results showed that "aesthetic appreciation" of green and blue spaces was the highest-ranking CT among the panellists. The study found that bogs, forests, cemeteries, and schoolyard gardens were highly rated for various subgroups of CTs, including "spiritual", "ornamental", and "educational values". Blue spaces such as the sea and bogs were also found to promote rest, stress relief, restoration, and calm. In contrast, vacant lots and quarries offer limited well-being benefits.

The correlation between cultural ecosystem services and well-being is significant, and their loss can have negative impacts on human health and well-being, human–ecological relationships, and social and economic values. Therefore, it is important to map and

assess cultural ecosystem services to identify areas for conservation and sustainable use. Additionally, cultural ecosystem services studies can help to measure the nonmaterial benefits humans receive from nature. Policymakers and planners can benefit from the insights provided by these studies to make more informed decisions about land use and management practices.

Filling a gap in current knowledge, this pilot study presents a suitable method for evaluating the relationship between cultural ecosystem services and well-being in a peri-urban area. While research-to-practice gaps and a lack of suitable methods exist in the application of cultural ecosystem services generally, this study provides valuable information on how to effectively manage and conserve these services, particularly in urbanised areas. Although this study needs further testing, its findings offer important insights into the correlation between cultural ecosystem services and well-being, highlighting the need for continued research and action to protect and sustain these vital resources.

However, it is necessary to acknowledge that our sample size was limited and the study was conducted in a specific geographic area for the reason of testing the novel method (RQ2). Additionally, the perception of cultural ecosystem services and well-being may vary among different groups of people, such as those of different age, gender, and socioeconomic status, among others. Future research should also consider including a more diverse group of experts and stakeholders with varying backgrounds and perspectives to ensure that the results are representative of a wider population. This can improve the validity and reliability of the study's conclusions and contribute to a more comprehensive understanding of the relationship between nature and well-being in peri-urban areas. While the study aimed to demonstrate the feasibility of evaluating perceptions rather than achieving complete consensus, the moderate level of agreement among the panellists supports the validity of the systematic approach used.

Lastly, we aim to enhance our findings by creating a comprehensive approach to assess a wider range of indicators within the matrix, including the crucial aspects of "duration" and "quantity" [59]. By incorporating these additional parameters, we can acquire a more profound comprehension of the amount of time spent in green and blue spaces, as well as the extent of their presence (including cultural ecosystem services) enabling us to identify and prioritise their contribution to well-being.

**Supplementary Materials:** The following supporting information can be downloaded at: https://www.mdpi.com/article/10.3390/su151310214/s1.

**Author Contributions:** Conceptualization, F.N. and M.K.; data curation, F.N., L.-M.T. and M.K.; formal analysis, F.N., L.-M.T. and M.K.; funding acquisition, F.N.; investigation, F.N., L.-M.T. and M.K.; methodology F.N., L.-M.T. and M.K.; project administration, F.N. and M.K., resources, F.N., M.K., J.S. and S.B.; software, F.N.; supervision, J.S. and M.K.; validation, J.S., M.K. and S.B.; visualization, F.N.; writing—original draft, F.N.; writing—review and editing, F.N., J.S., M.K. and S.B. All authors have read and agreed to the published version of the manuscript.

**Funding:** This research was funded by the Estonian Government Scholarship for Doctoral Students.

**Institutional Review Board Statement:** Not applicable.

**Informed Consent Statement:** Not applicable.

**Data Availability Statement:** Not applicable.

**Acknowledgments:** We would like to thank the Harku municipality experts who graciously shared their time and expertise during the interviews. We would like to extend our sincere gratitude to our colleagues Martti Veldi, for his contribution to generating the map for this study (Figure 1), and Peeter Vassiljev for his advice on data interpretation. Finally, we express our sincere gratitude for the time and effort the reviewers have invested in reviewing our manuscript. Their feedback and expertise were invaluable in improving the quality of our work, we appreciate their detailed comments and suggestions.

**Conflicts of Interest:** The authors declare no conflict of interest.

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
