# Peer review of "Assessment of Cultural Ecosystem Services and Well-Being: Testing a Method for Evaluating Natural Environment and Contact Types in the Harku Municipality, Estonia"

_sustainability, doi:10.3390/su151310214_

Round 1

Reviewer 1 Report (Previous Reviewer 2)

Based on the revision format and response to the reviewer cover letter, recommended that manuscript entitled "Assessment of Cultural Ecosystem Services and Well-being: Testing a method for evaluating natural environment and contact types in the Harku Municipality, Estonia" for publication in Sustainability.

Minor editing of English language required

Author Response

Reviewer 2 Report (Previous Reviewer 3)

[Sustainability] Manuscript ID: sustainability-2446746

GENERAL COMMENTS

The revised manuscript has shown improvement, and I would like to thank the authors for submitting a more comprehensive version. Including a comparison of agreement among experts is a valuable addition, as it sheds light on the variations in suitable combinations of Nature Exposure Therapies (NETs) and Cognitive Therapies (CTs). This finding serves as an important foundation for further investigations, especially among different target groups. The results of the study reveal interesting patterns regarding the impact of different NETs and CTs on well-being.

SPECIFIC COMMENTS

Point 1, L181: To enhance understanding and facilitate comparison, it would be beneficial to include a scale in Figure 2, also allowing for a clearer depiction of the almost sprawl-like effect.

Supplementary file: It is unclear what data the last column represents. Clarification on this matter would be helpful.

SUGGESTIONS:

· Conducting additional investigations involving various target groups is strongly supported. For instance, the weaker agreement among experts in the case of "the green landscape elements and education" could be a topic of debate, as correctly mentioned in the Discussion. Generally, the green landscape and its elements are valuable sources of information. If appropriately interpreted, they can offer valuable opportunities for the educational process. Furthermore, it is important not to overlook the well-being benefits associated with outdoor experiences.

· It is highly recommended to use the same number of decimal places for all numbers. Consistency in the manuscript text, figures, and supplementary table would enhance clarity and precision.

Author Response

This manuscript is a resubmission of an earlier submission. The following is a list of the peer review reports and author responses from that submission.

Round 1

Reviewer 1 Report

This work deals with a topic that, although much explored in the literature, is in any case very important: the function of ecosystems and well-being. It is also very interesting to understand how ecosystems are used in a country that is generally little known and has gone through major political and social changes in recent decades. The article as a whole would be balanced as a structure and clearly written, but there are major problems especially in the methodological part in relation to the sample considered.

The sample n=5 is too small, although you interviewed 5 “experts” you cannot work with such a small number of interviews/questionnaires.  Moreover, you do not explain how these experts can know people's perceptions/use or attitude in relation to all the ecosystems considered, how do they know? Do they have statistical data? On what do they base their answers? This is not clear.

In the results, you evaluate just the average mean score, but is it the mean statistically significant? Normally, to prove the consistency of a result, a statistical test is needed. What significance can have a score of 2,6 in comparison with 2,8?  For instance what can mean “it scored moderately high (2.4)”  if you do not show a statistic test?  It is not possible to do any kind of robust quantitative study -as appears to be your intent- with a such sample and especially if we are talking about a municipality of 17,364 inhabitants. 

Therefore, from my point of view the work cannot be accepted, the sample must be increased or instead of experts, the survey could directly address to people, in this case, one could consider demographic data such as age, origin, social class, educational qualification, type of work, etc.

Furthermore, the introduction can be improved by considering other works, for instance see the references below. Indeed, one could consider ecosystems not only functional for human well-being but having an importance in themselves for the sake of biodiversity and other nonhuman beings. This last part could be explored considering the recent shift of paradigms in anthropological and social sciences: e.g. ontological turn, animal turn.

Agarwala, M., Atkinson, G., Fry, B. P., Homewood, K., Mourato, S., Rowcliffe, J. M., ... & Milner-Gulland, E. J. (2014). Assessing the relationship between human well-being and ecosystem services: a review of frameworks. Conservation and Society12(4), 437-449.

Bennett, E. M., Cramer, W., Begossi, A., Cundill, G., Díaz, S., Egoh, B. N., ... & Woodward, G. (2015). Linking biodiversity, ecosystem services, and human well-being: three challenges for designing research for sustainability. Current opinion in environmental sustainability14, 76-85.

Sandifer, P. A., Sutton-Grier, A. E., & Ward, B. P. (2015). Exploring connections among nature, biodiversity, ecosystem services, and human health and well-being: Opportunities to enhance health and biodiversity conservation. Ecosystem services12, 1-15.

Reviewer 2 Report

I just Review the manuscript entitled"  Cultural ecosystem services and well-being assessment: 2 natural environment and contact types in the Harku 3 municipality, Estonia"

The subject is really interesting and would be important regarding the relationship between CES and W.B.

HOWEVER, WOULD BE BETTER TO REVISE IN SOME PARTS AS FOLLOWS:

1- IN THE LITERATURE REVIEW THE LACK OF DEFINITIONS AND THE IMPORTANCE OF CES APPLICATION IN URBAN LANDSCAPE AND PLANNING WOULD BE CLARIFIED AND WOULD BE BETTER TO CITE MORE REFERENCES.

2- IN THE METHODOLOGY AUTHOR JUST SELECTED 5 EXPERTS THAT WOULD NOT BE VALID FOR THE INTERVIEWERS, PLEASE DESCRIBE THE WAY THAT JUST FOCUSES ON THESE 5 EXPERTS. 

3-CONCLUSION WOULD BE MORE DESCIBED AND IM[PROVED REGARDING THE CORALTION BETWEEN W.B AND CES.

Reviewer 3 Report

[Sustainability] Manuscript ID: sustainability-2192533

The presented paper presents a novel methodology for evaluating cultural ecosystem services (CES) and how they contribute to subjective well-being (WB), using the assessment of natural environment types (NETs) and contact types (CTs). Experts interview was conducted and resulted in a matrix of averaged mean scores for each NETs and CTs.

GENERAL COMMENTS

1. The manuscript represents a country study. Could some of its results and conclusions be generalized on the (Central)European level?

2. Authors for evaluating the CES and how these contribute to subjective WB interviewed five municipal experts from different fields. For further research and evaluation, the suggestions would be to broaden this group of experts and include other target groups benefiting from public spaces (green and/or blue), e.g., families with children, elderly, etc., and reach the statistically relevant number of respondents (obtaining thus information also on preferences of various age groups and/or their type of occupation, etc.).

3. Starting from Figure 5 all charts should bear clear descriptions, including those of x- and y-axes.

SPECIFIC COMMENTS:

P2, L93: I suggest explaining CTs and NETs abbreviations, as they are used in the text for the first time.

P7, L253: Identical subtitle as in P7, L243, shouldn’t it be “Psychological”?

P9, L291: Correct the number of the Figure (Figure 1 to Figure 6).

P10, L302: Correct the number of the Figure (Figure 2 to Figure 8).

P10, L312: Correct the number of the Figure (Figure 3 to Figure 9).

P11, above L318: Correct the number of the Figure (Figure 4 to Figure 10).

P11, L333: Duplicity; the research question has already been formulated (see P2, L93).

P13, L397: The chart legend is not complete (the chart has 5 categories, but the legend shows only 3). 

P14, L434: Correct the number of the Figure (Figure 5 to Figure 12).

P14, L444: Correct the number of the Figure (Figure 6 to Figure 13).

P15, L450: Missing space between the bracket and reference.

P15, L471: Reference to Figure 14 that is not present in the manuscript.

P15, between L471 and L472: Figure title (Figure 7) with no figure.

Reviewer 4 Report

GENERAL COMMENTS:

Thank you for your manuscript, the purpose of which I understand is to develop and test a method for evaluating cultural ecosystem services in relation to well-being. The authors proposed an interesting framework for evaluating CES that considers different types of contact with the natural environment, offering a better understanding of CES that covers a broader range of the different aspects that characterize CES and that have been generally addressed in the literature in isolation and not jointly as proposed by the authors. In the same way, this proposal offers interesting elements to expand the matrix proposed by Burkhard et al (2014). Additionally, the authors delve into the conceptualization of well-being and its link with the CES, which is a valuable and novel proposal.

In general, the results obtained by the authors are interesting. My comments on the different parts of the manuscript can be found below:

SPECIFIC COMMENTS:

The subheading in section 4.1 (line 319) is unnecessary and confusing, I would recommend removing it and developing the idea in the discussion directly without the subheading “Introduction” and starting 4.1 on what would be 4.2.

The results follow a similar order to that of the general matrix with the heat map, however, in the discussion this order changes, and instead of following the same order to describe the type of Restorative, Social, and Cognitive contact, it is passed directly to the subtype. While this order seems to answer the research question, it also seems to hide the other types of contact. In this sense, I would recommend including at least one paragraph that refers to the type of contact (Restorative, Social, and Cognitive) indicating in what type of natural environment they are dominant and why. Or another alternative that could improve the presentation of the discussion would be to follow the same order of the results in which for each type of main contact (Social, Restorative, and Cognitive) are included which of the contact subtypes were the ones with the highest and lowest value since from my point of view this is the most valuable aspect of the investigation.
